# Public Health Risk Evaluation through Mathematical Optimization in the Process of PPPs

**DOI:** 10.3390/ijerph20021175

**Published:** 2023-01-09

**Authors:** Mohammad Heydari, Kin Keung Lai, Victor Shi, Feng Xiao

**Affiliations:** 1Business College, Southwest University, Chongqing 400715, China; 2International Business School, Shaanxi Normal University, Xi’an 710119, China; 3Lazaridis School of Business & Economics, Wilfrid Laurier University, Waterloo, ON N2L3C5, Canada

**Keywords:** public health, public–private partnerships (PPPs), private sector, public sector, mathematical modeling

## Abstract

The public sector is becoming increasingly appealing. In the context of declining public money to support health studies and public health interventions, public–private partnerships with entities (including government agencies and scientific research institutes) are becoming increasingly important. When forming this type of cooperation, the participants highlight synergies between the private partners and the public’s missions or goals. The tasks of private and public sector actors, on the other hand, frequently diverge significantly. The integrity and honesty of public officials, institutions, trust, and faith in those individuals and institutions may all be jeopardized by these collaborations. In this study, we use the institutional corruption framework to highlight systemic concerns raised by PPPs affiliated with the governments of one of South Asia’s countries. Overall analytical frameworks for such collaborations tend to downplay or disregard these systemic impacts and their ethical implications, as we argue. We offer some guidelines for public sector stakeholders that want to think about PPPs in a more systemic and analytical way. Partnership as a default paradigm for engagement with the private sector needs to be reconsidered by public sector participants. They also need to be more vocal about which goals they can and cannot fulfill, given the limitations of public financing resources.

## 1. Introduction

Corruption has infiltrated the public health system. As researchers, authors, public health workers, and health ministers, we have witnessed widespread dishonesty and deception. Despite the fact that corruption is one of the most serious roadblocks to universal healthcare adoption around the world, it is rarely discussed publicly. The current article discusses systematic literature reviews (SLRs) on the size of public health corruption as a barrier to long-term growth, as well as how it began and what is happening currently. It also highlights people’s anxieties about the topic, what is needed to combat corruption, and the academic and research communities’ accountability in all countries, regardless of their economic development level. Scholars, politicians, and funders must consider corruption as a serious problem in the same way they consider diseases. If we are to accomplish the Sustainable Development Goals and ensure that everyone lives in safety, global health corruption must no longer be an open secret.

Inflection is used to convey the meaning of this interrogative. It questions, perhaps skeptically or befuddlingly, what objections there might be to PPPs aimed at addressing some of our most important public health concerns, from the mouths of proponents of PPPs related to food and health. This is due, in part, to the way such partnerships are frequently described by participants and proponents: they are a “win-win-win” situation for the public sector actor, private sector players, and the general public or relevant public (usually classified or reframed as consumers). The question cynically suggests a quid pro quo—that the private partner has acquired something meaningful in exchange for the public partner’s support, and that the public partner has been “used” or compromised as a result of this interaction, according to critics of PPPs. However, the ethical implications of PPPs are more complex than either of these viewpoints suggest [1,2].

Applying the framework of institutional corruption, we draw out some of these implications, concentrating on PPPs’ potential systemic impacts—consequences usually ignored in the academic literature and, even more so, in policy discussions [2,3]. These systemic impacts tend to be insidious and contain the erosion of public institutions’ tasks, integrity, trust, and confidence in those institutions. Building on these concerns, we draw attention to the limitations of prevailing analytical strategies to the ethics of public–private partnerships and offer alternative ways to address the systemic ethical problems they raise.

Before this evaluation, we provide a brief description and taxonomy of public–private partnerships and an overview of institutional corruption as an analytical framework or lens for public–private partnerships’ ethical analysis. This manuscript focuses on two research questions:Q1: What aspects of the project encourage corruption?

This first query is required to determine whether there are characteristics that increase the likelihood of corruption in the projects. It is imperative to know the answer to this issue, especially for policymakers in corrupt nations. Let us take the example of a “*functional purpose*” (such as providing a specific amount of electricity in a specific area) that can be met by two alternative projects, type *A* and *B*, and that one of them (*say*, *B*) is more likely to be corrupted than the other. Therefore, in a corrupt project scenario, A ought to be the best option in accordance with this criterion.

Q2: How does project performance suffer from a corrupt context?

This topic intends to emphasize the impact of corruption by comparing identical projects in different nations because projects may perform poorly in terms of schedule and budget even in “*non-corrupt countries*”.

A critical literary analysis of sources, the majority outside the field of project management, is used to respond to *question 1. The second question* is addressed by the literature review and further investigated with a case study in (Section 4, Section 4.1 and Section 4.2). First, the evaluation of the literature helps us to codify the major concepts that are important for this study, such as corruption, project context, corrupt project context, megaproject, etc. Second, it enables a list of key drivers indicating the kind of projects that are more likely to entail corruption to be included in the *RQ1 response*. Thirdly, it describes the impact corruption-related projects have on their performance over the course of their existence. In conclusion, the literature study emphasizes two key points: the causes of corruption and how it affects project outcomes.

Although corruption is frequently found in megaprojects, even in nations with low levels of corruption, megaprojects are frequently linked to subpar performance. As a result, the technique analyzes the large-scale high-speed rail systems in Europe and around the world. The comparison takes into account two key points of view: first, the project circumstances and the degree of corruption and second, the performance of the megaprojects was standardized and adjusted to take into account various technological, urbanistic, and environmental factors. The case study (Section 4, Section 4.1 and Section 4.2) of the Italian high-speed rail development serves as the foundation for the comparison [4]. Because it is supplied in a corrupt project setting and is technologically comparable to the other European high-speed rail initiatives, this example is considered a model. The case study’s goal is to demonstrate how the pervasive problem of corruption and poor project performance are related; this strategy is put into practice in accordance with the guidelines provided by [5,6]. The project background, the longitudinal view of the project lifecycle, and the transversal view of the project performance make up the three main views of the case study. Because it is challenging to clearly demonstrate the presence of corruption in projects and because the research is not focused on specific instances of corruption but rather on projects that are delivered in a “*corrupt project context*”, the case study is created in this manner. By mentioning the project setting and displaying the outcomes of the legal proceedings and investigations connected to the project, the document subtly illustrates the existence of corruption. Figure 1 summarises the two RQs, the main research constructs, and their haphazard connections.

This publication, in particular, demonstrates how project and project management performance changes over the project lifetime by utilizing the research paradigm from [8]. Five criteria are used in Merrow’s methodology to assess the performance of the megaproject. Each parameter has a threshold value linked with it, making it possible to determine whether the performance is sufficient or not (Table 1).

## 2. What Is a Public–Private Partnership?

A PPP does not have a standard definition. PPPs are defined by the World Health Organization (WHO) as “a wide range of enterprises involving a diversity of arrangements, varied concerning participation, legal status, governance, management, policy-setting prerogatives, contributions, and operational tasks” [9]. The term public partner is also not uniformly defined. It is used in this paper to include a government agency or official, an academic research institution, a nongovernmental organization, and an international nongovernmental organization, among others. The term private partner is used here to include corporations, trade associations, and other organizations representing industry interests. For my analysis, an organization that is substantially dependent on the industry’s financial contributions to conduct its operations should also be considered a private partner, absent a particularly compelling case that it should be regarded as otherwise. Entities that depend on both industry funding and public sector actors’ participation may themselves be considered PPPs.

### Taxonomies of Public–Private Partnerships

PPPs are classified in a number of ways. The United Nations Standing Committee on Nutrition (UNSCN), for example, divides partnerships into four categories: (1) direct finance, (2) in-kind contributions (which might include products and services), (3) conversation (which includes information exchange), and (4) collaborative delivery. The UNSCN’s 2006-7 policy, for example, declares that cash and in-kind contributions from industry are “*off the table*”. The policy states that [i]n order to avoid institutional conflict of interest, the Steering Committee will ensure that the nation’s standing committee on nutrition does not approve financial or in-kind contributions from food-related PSOs [private sector organizations] for any of the organization’s activities, whether they are developed through Working Groups or the Steering Committee/Secretariat-based work plans. Direct funding and in-kind contributions for non-food PSOs can only come from PSOs that have received “*acceptable analysis evaluations about their performance on human rights, labor rights, environment, and good governance standards* [10]”.

Other classification systems focus on the relevance of the PPP to the partners’ missions rather than on the partners’ product or the nature of the contribution. For example, James Austin and Vivica Kraak, and their co-authors, suggest that philanthropic partnerships—often single gifts—involve low levels of engagement between the parties, are “*peripheral*” to the tasks of the parties, and are of “*minor*” strategic value [11]. They contrast a second phase or level of partnership, which both papers term “*transactional*”. Such relationships are described as involving more significant mutual benefits than a philanthropic partnership. The third phase or level is termed “*integration*” by Austin and described—less transparently—as “*transformational*” by Kraak et al. A central feature of Austin’s integrative partnerships is “*mission mesh*”. Given that the tasks of the public and private partners inevitably diverge in meaningful ways, for example, when the public partner is a government agency with a regulatory mandate that encompasses the activities of the private partner, the meshing of the missions of public and private partners raises significant ethical concerns that will be discussed further in this paper.

Philanthropic gifts may be much more central to both parties’ missions and have more excellent significant strategic value than Austin and Kraak appear to recognize. This may be illustrated by the USD 10 million gift from a foundation established by the American Beverage Association to the Children’s Hospital of Philadelphia (CHOP) in 2011 to fund clinical care, policy studies, outreach, and prevention efforts for childhood obesity [12]. The donation was pledged while Philadelphia’s City Council considered a resolution that would introduce a 2 cents per ounce tax on sugar-sweetened beverages [13]. Shortly after that, the answer was defeated [14]. (The implications of this are discussed below). Other less dramatic examples include smaller-scale philanthropy from industry and trade associations to support the construction of playgrounds and other child-friendly outdoor spaces in inner cities.

## 3. PPPs and Corruption: “*before*” and “*after*” the Illegal Act

This section briefly reviews the characteristics of PPPs [15] and their relationship to corruption. It highlights how studies have focused mainly on analyzing the risks of corruption in each phase of the PPP cycle and the preventive measures to deter illegal acts. However, recent experience shows that the number of these crimes detected in infrastructure projects has not diminished despite mounting efforts essential to combat them. On the contrary, as a consequence of these efforts and other events, corruption cases have been discovered in connection with public works even more. In this scenario, it is no longer relevant to address how the institutional framework operates “*before*” the illegal act but also “*after*” the illegality is committed. For this reason, the concentrate is placed on the need to study the effects of corruption on PPPs once the preventive measures on transparency and integrity have failed to prevent illegal activity [16]. In particular, this work highlights how the absence of mechanisms and institutional frameworks for adequate regulation to sanction corruption in PPPs brings about effects that are especially harmful to the sector.

### 3.1. PPPs and Corruption: A Brief Overview of the Existing Literature

PPPs are long-term contracts between a private sector and a government entity to provide a public asset or service. The private section bears significant risk and management liability, and in which remuneration is connected to performance [17,18]. The importance of PPP contracts has grown steadily over time, both in developed and emerging countries, but particularly in the latter, as of the 1990s, as shown in Figure 2.

The pressing need for investment in infrastructure and the growing limitations faced by countries in fiscal space to finance traditional public works have made PPPs an attractive mechanism for many governments. PPP contracts confer added value if compared to conventional public procurement when there exists an appropriate allocation of risks and benefits among the public and the private parties. In this sense, the public sector assumes those risks that it is better qualified to control, mitigate, or bear while transferring the remaining risks to the private party.

In general, the most relevant risks of any project are divided into two phases: the design and construction level, and the operation and maintenance phase. Once the risk has been identified, it is determined which party is in a better position to control or take on this risk. For instance, during the construction stage, the private sector must comply with environmental protection regulations and the measures recommended in the corresponding environmental impact assessments, adjusting their processes and construction methods so that their work is in compliance with the applicable legislation. Similarly, throughout the operation and maintenance phase, the public sector should ensure periodic update mechanisms in line with current inflation indicators, so as to avoid cost overruns in the operation and maintenance of the infrastructure.

The ability of the public party to handle complicated contracts, as well as the economic efficiencies given by the private sector through improved technology and incentives to develop and run infrastructure under a PPP model, all contribute to the PPP model’s worth. In a principal–agent setting, Sherstha et al. [19,20,21,22] proposed the idea of a balanced risk allocation mannequin where both the government and the concessionaire try to achieve their goals to improve their utility level (Figure 3). Taking on risk accountabilities in a balanced manner can result in a successful collaboration for both parties [19,23,24]. However, if one party has an information advantage, it may try to increase its utility expense to its partner. When this happens, the balance is upset, and the chances of completing the job are reduced [25]. As a result, PPP agreements should be written and negotiated in such a way that encourages both sectors to achieve a win–win balance by efficiently allocating risks and responsibilities.

Undoubtedly, the institutional framework that regulates PPPs is a crucial element for the proper alignment of incentives for each of the parties involved (ministries, entities, banks, and companies, among others) throughout each phase of the project cycle (design, approval, bidding, selection, contracting, monitoring, and control). Most countries in the region have developed legal and institutional frameworks to regulate the different levels of the project cycle, adapting them to the size of their economy and their degree of decentralization [26].

The literature on PPPs and corruption has focused mainly on analyzing how this type of contract’s institutional and contractual characteristics differ from those of traditional public works procurement in terms of their capacity to prevent illegal acts throughout the different stages of the project cycle. Contracts entered into the public sector are generally more advantageous than those entered into by private parties, and they pose specific risks of fraud and corruption [27]. Some argue that the risks of crime in this type of contract could be even more significant [28]. This is because PPP contracts entail specific risks inherent to their very nature [29,30]. These risks derive from the contracts’ flexibility and a high degree of indeterminacy, mainly since they are incomplete, long-term contracts, which makes it impossible to foresee all the circumstances that could unfold throughout the lifecycle of the contractual relationship. Similarly, PPPs may present specific institutional risks, such as those caused by the lack of agencies that specialize in public–private relations or those caused by weak control tools or defects in the mechanisms to ensure transparency and access to information [31].

In this context, PPP projects encompass at least four stages that demand specific anti-corruption measures: (i) the decision to choose a PPP instead of one of the traditional models for the execution of a project; (ii) the design, drafting, and approval of the project; (iii) the contractor selection process and the awarding of the PPP contract; and (iv) the execution, management, supervision, and termination of the contractual relationship. In this regard, it is thought necessary to implement controls during the pre-bidding stage to prevent acts of corruption aimed at manipulating risk transfer or overestimating the private party’s efficiency in order to promote inconvenient PPPs.

This has led some authors to advise against the transfer of risks and the bundling of building, operation, and maintenance phases in contexts of high corruption of institutional weakness [32]. Other opinions aim to guarantee the application of robust public–private comparator methodologies and increase transparency during the pre-bidding phase [33], including the preparation, development, and approval stage of the project [34].

Although the literature on this topic tends to show the existence of certain margins of discretion to manipulate any stage of the bidding phase, transparent, competitive bidding procedures, with adequate and reasonable timeframes, have asserted themselves as the most appropriate mechanism to reduce opportunities for corruption during the contractor selection process [34,35,36]. This has, for instance, prompted some international organizations to issue best practices documents for public procurement in PPP projects [25].

According to Guasch, “[r]arely has a government denied an addendum request if the consequence of doing so was the cancellation of the project or the termination of the contract. Only 3% of PPP projects in the world have been canceled. This leads to an inescapable rationale, on the part of the private operator, to trust that the government will accept its requests for addenda, especially if the system or officials with decision-making power have been bought off with bribes” [37]. To prevent this situation, legal limits have been imposed on the percentage of the contract value renegotiated. Special authorizations have been required to allow the signing of appendices [38]. Similarly, the inclusion of contingency clauses has been proposed, for example, to extend the duration of the contracts in a pre-determined manner when facing exogenous shocks, such as those derived from changes in traffic flows or verifiable project costs, among others [39]. Figure 4 illustrates the different phases of a PPP project, identifying some of the corruption risks in each of them.

As can be seen, the literature on corruption and PPPs focuses on specific aspects of this type of contract, which, via its very nature, poses particular risks in the event of a breakdown. The authors focus on the rules that can dissuade and prevent crime by introducing different mechanisms to ensure transparency and monitor the project [40]. Responses have also been aimed at strengthening institutional frameworks and agencies or units specializing in PPPs, which involve implementing mechanisms to guarantee, among other issues. These technical and human resources minimize corruption (robust value-for-money analysis, standardization of contracts and procedures, project monitoring, etc.).

The competencies and authority of specialized units dealing with PPPs may vary from one jurisdiction to another, depending on the institutional framework in which they operate. However, they share the general objective of establishing a permanent government structure with the knowledge required to identify and manage the opportunities for linking the public and private sectors [41]. These units are complemented by other government agencies that provide advisory, operational, and technical support throughout the different stages of the project. For this reason, in many countries, these specialized units play an important role before, during, and after the tender process. However, at present, their specific role in dealing with corruption in PPPs is not clearly defined. Nonetheless, these government agencies contribute indirectly to the prevention of crime through the correct exercise of their competencies, aimed at ensuring the proper utilization of PPPs under formally established principles of transparency and integrity, but without a control and monitoring function. This function is exercised by those explicitly created for that purpose, namely anti-corruption offices or the corresponding comptroller’s agencies.

### 3.2. New Approach to PPPs and Corruption

The detection of corrupt activities in public works has enhanced in recent years, partly thanks to the improved effectiveness of the controls implemented since the 1990s [42]. This new scenario requires an analysis of the economic effects of the existing legal responses for corrupt activities, particularly concerning the paralysis of the works and investments and the damages caused to third parties unrelated to the illicit act, including those interested in the project [43]. This paper is not intended to analyze the core causes of corruption in public works and the construction sector (On this matter, please refer to Matthews (2016), Wells (2015), Transparency International (2006), Kenny (2006), and Susan Rose-Ackerman (1975), among others [44,45,46,47,48]), which may derive from systemic or structural nature (economic informality and institutional weakness, among others).

What is examined is how the primary legal mechanism to deal with corruption interplays with the specific characteristics of PPPs, bringing about particularly pernicious effects that have prompted some countries in the region to readjust their legal frameworks. On the other hand, the object of analysis is how the potential annulment of an ongoing contract triggers dire consequences in contractual structures based on structured financing and “*contractual cascades*” that combine public and private law. Figure 5 demonstrates a basic overview of the primary contractual relationships established in a typical PPP contract.

The PPP (or concession) agreement, which is entered into by the government and the contractor to set, among other things, construction commitments and quality standards, is the most important contractual component of the PPP system. This also creates the flow of funds (tolls, fees, or availability payments, among others) that will be used to repay the project’s financing and distribute dividends among the shareholders. This constituent factor or “*mother contract*” is governed by public law, specifically by the PPP law, and by general administrative law [49]. Thus, PPP contracts usually consist of a series of consecutive documents that include the terms and conditions of the tender or bidding process, the corresponding clarification notices, the offer of the successful bidder, the award’s administrative act, and the contract itself and its subsequent addenda.

Under administrative law, upon verification of an act of corruption, the executive decision that initially conferred validity on the PPP contract is rendered null and void. Such effect, in principle, also entails generating ineffective any other contract directly or indirectly associated with it. The act of corruption triggers the contract’s annulment as a legal sanction that extinguishes the expected effects of the PPP contract [50,51]. Therefore, as the contract is rendered null and void, in principle, parties must restitute any assets they might have received or, if this were not possible, pay back their value, in which case the contractor shall have no further right unless this resulted in the unjust impoverishment of the contractor and the unjust enrichment of the state. However, this restitution or compensation does not operate if the parties are responsible for the invalidation, resulting in corruption cases. Corruption and fraud are contrary to the principle of good faith and, consequently, contrary to any legal system. It would be contradictory to allow the legal system to protect rights resulting from illegal activities.

Because contract annulment is a reaction authorized by administrative law, and because there are no particular PPP rules, financiers and contractors see nullity as the most likely result of corrupt acts. In this sense, there are two characteristics inherent to PPP projects that render the results of a potential declaration of nullity of contracts particularly detrimental. These help explain why PPPs in some Latin America and the Caribbean (LAC) countries have become paralyzed as a result of acts of corruption and the need for legislative reform detailed in the following section.

PPPs have the distinct economic and financial aspects of being an investment that is recouped over time from the flow of revenues created by the project itself (tolls, fees, availability payments, and others). Under this structure, when facing a credit event or an early termination of the relationship, there is no available recourse to the corporate balance sheet of any of the companies that constitute the “*project company*” (or particular purpose vehicle), *[PPPs are based on a structured financing project in which an SPV owned by a consortium of companies, which may include the participation of the public sector, which sign and execute the contract. This* “*project company*” *will then be in charge of the development, construction, maintenance, and operation of the works throughout the duration of the contractual relationship, outsourcing, in turn, the goods and services necessary to meet the obligations under the contract]*. However, rather exclusively to the flow of funds arising from the project, the assets were derived from investments made, and other rights and obligations contemplated in the PPP contract. As a result, unlike traditional public works procurement, where investment payback occurs as the work is completed and upon public authorities’ recognition that the work has been completed, investment payback in PPP projects occurs over a longer period (usually greater than 15 or 20 years) through the flow of funds generated by the infrastructure and through government contributions known as availability payments, which result from adherence to the governing documents.

Therefore, financiers are highly vulnerable to the nullity of contract as a response to an act of corruption, given that nullity eliminates the source of repayment, which may even occur after the entire investment has already been made. There is also no recourse to the company’s financial position in PPPs but rather to that of the project company. In contrast, in traditional public works, the nullity of the contract does not prevent the financier from gaining access to the construction company’s corporate balance sheet [52]. Similarly, the investment payback in traditional public works does not occur over time but upon acknowledgment of completion of each of the stages into which the work is divided, thus reducing the amount of investment at risk. These characteristics constitute the main reason for the paralysis of PPP financing in some countries in the region due to recent corruption problems (refer to Céspedes (2017), La República (2017), Tola (2017), and Semana (2017a, b); [53,54,55,56,57]) and the legal reforms promoted by governments to solve said difficulties.

From a legal and contractual standpoint, PPPs’ characteristic feature is the “*contractual cascade*” that emerges from the public–private collaboration, combining aspects of administrative and commercial law. Stemming from the public–private liaison, a series of legal relations begin to take shape, involving a large, indeterminate, and varying number of players and contracts over a long period. This means that a large number of agreements are required to carry out the entire project, all of which are dependent on the main, or mother, contract, including the loan agreement with the banking system and capital markets, the construction contract with the building contractor, the operation and maintenance contracts with service providers (operators), shareholder and equity contribution agreements for the project company, and the insurance contract. Thus, for example, the construction company outsources many tasks to other companies, and so forth. Likewise, throughout the project’s extensive duration and the public–private collaboration, the parties involved in each of the referred legal relations and contracts may vary. For instance, once the construction phase is concluded, the initial industrial partners can divest their participation in the project; financiers can sell their credit. The project might be refinanced with other banks or through operations in the capital markets.

Hence, when facing an event that results in the nullity of an existing PPP contract, many links in the public–private contractual chain are broken, which are increasingly distant from the “*mother contract*” and, over time, also from the parties that initially composed it. Thus, the consequences of nullity have a “*trickle-down effect*” along the contractual chain, which extends far beyond the origin of the wrongful act, as shown in Figure 6.

It shall be noted that even though the inclusion of anti-corruption contract clauses is positive and necessary, it is also true that these provisions are not sufficient to solve the problem.

Any contractual stipulation that seeks to regulate the effects of corrupt activities must necessarily be consistent with the annulment regime’s general provisions. If an annulment is the only sanction contemplated in the existing general legislation, its grounds, effects, and consequences cannot be modified, negotiated, or agreed upon in a contract, as they fall under the scope of public order provisions that cannot be waived or modified by the parties [58].

Nonetheless, the inclusion of anti-corruption clauses in public contracts is becoming increasingly common and constitutes an additional deterrent for the commission of illegal acts.

Multilateral development organizations also include such provisions in their loan agreements. The IDB, for example, incorporates a “Prohibited Practices” clause that includes acts of corruption, fraud, coercion, or collusion, among others. The occurrence of a prohibited practice enables the bank to take a number of measures, including denying financing of contracts, declaring the ineligibility of a procurement process or a company to participate in activities financed by the bank, and imposing other sanctions, such as fines.

These clauses generally include three components: (i) a representation and warranty of the contractor representing and warranting that no bribes have been paid either directly or indirectly nor corruption acts have been committed during the period of contract formation; (ii) the obligation to conduct business with transparency, honesty, and integrity, which, in turn, implies the prohibition for the contractor, partners, controller’s agents, and employees to commit corrupt acts during the term of the contract (to prevent acts of corruption, the parties may agree to the obligation of establishing internal compliance corporate programs and codes of conduct); and (iii) the obligation to promptly report to the competent authorities any act of corruption that might come to their knowledge [59,60]. The most common consequence for non-compliance included in contracts is the unilateral termination of the agreement without the right to compensation, including the imposition of penalties on the breaching party.

However, as stated above, these contractual provisions come into conflict with annulment provisions in the governing law for PPP contracts. This means that the parties to an agreement cannot “*convert*” the act of corruption into a requirement for early termination of the contractual relationship since disintegration triggers a penalty (the nullity of the contract), which, as such, extinguishes all the provisions of the agreement and its effects. Therefore, nullity cannot be construed as an event of early termination as it is inherently different from this category, which can be mutually agreed upon by the parties.

Common law contracts, such as those executed in the United Kingdom, include clauses that provide for the early termination of the contract on the basis of delivery of corrupt gifts and the commission of fraud (termination on corrupt gifts and fraud). The provision seeks to strike a balance between the public sector’s interest in terminating its contractual relationship with a corrupt partner and the interest of third parties, such as financiers who have not taken part in any transaction involving prohibited activities. Contracts specify which actions constitute corrupt gifts and fraud. If the private sector is accountable for the prohibited action, the public sector has the right to terminate the contract upon payment of outstanding financial obligations. In addition, the government must be compensated and given all the project assets (see [61]).

The distinction is significant for two reasons. First, early termination involves payment to the contractor of some sort of compensation following the valuation methodology and calculation procedure established for that purpose. Contrarily, a contract’s nullity resulting from a corrupt act does not allow for payment, as it punishes the parties’ bad faith.

Second, legislation usually contemplates a procedure to be followed when facing an early termination event. It would typically include the possibility of calling a new tender, summoning the bidders that participated in the tender process to complete the contract, or concluding the works through direct government management, among other options. In the case of the PPP contract for Improvements to the National Energy Security and Development of the South Peruvian Gas Pipeline, for example, Clause 21 of the contract empowers the grantor to appoint a concessionaire’s auditor to supervise the management of the service. The concessionaire must maintain the continuity of the service for a period of one year or until the concessionaire is replaced. In the event that this period elapses and there is no replacement, the concessionaire’s auditor shall take over the operation of the system until it is awarded to the new concessionaire. In turn, the grantor must call a new public auction for the transfer of the concession and handing over of the assets to the new concessionaire, and must pay the concessionaire a maximum amount equivalent to the book value of the goods under concession. In the absence of specific regulation on the consequences of nullity, the government can only resort, by analogy or extension and to the extent possible, to the exact mechanisms established for contract termination in an attempt to seek the rapid continuity of the paralyzed works, which will not always be achieved.

## 4. Evaluating Public–Private Partnership (PPP) Projects

PPPs are creative ways for the government to contract with the private sector, which contributes to their ability and capital to accomplish projects on time and on budget. The public sector, on the other hand, retains the responsibility to provide these services to the general public in order to benefit the people and produce economic growth and improved life quality [62]. PPPs are gaining favor around the world because they may effectively avoid the often harmful consequences of either exclusive public ownership and delivery services or outright privatization on the one hand. Furthermore, PPPs combine the best of both worlds: the public sector’s regulatory measures and public interest protection, and the private sector’s resources, management abilities, and technology. The fundamental goals of PPPs are to finance, develop, execute, and operate public party facilities and services. The following are some of the most important classifications associated with PPPs:(1)Long-term planning (generally up to thirty years) provisions for service;(2)Risk is transferred to private parties;(3)Different types of long-term contracts are drafted between legal entities and government agencies.

Although the types of public–private partnerships vary depending on the demands of governments for infrastructure services, there are two major categories of PPPs: institutionalized PPPs, which refer to all forms of joint ventures between public and private parties, and contractual PPPs. PPPs, in particular, can take numerous forms and include some or all of the following qualities [63,64]:(1)In general, the public party transfers facilities managed through itself to the private sector (with or without compensation) for the duration of the arrangement;(2)A facility is built, extended, or renovated by a private party;(3)The public party sets the facility’s operating characteristics;(4)Services are offered by a private entity that rents the facility for a set amount of time (generally via restrictions on operations and pricing);(5)At the conclusion of the agreement, the private party agreed to transfer the facility to the public party (with or without payment).

In the presence of hundreds of PPP projects’ applications, managers’ fundamental decision is to evaluate them and then select appropriate projects to implement. Motivated by the observations that evaluating PPP projects consist of multiple criteria, this paper proposes a decision approach that minimizes the total deviation from the best point to determine the weights connected via each measure.

The rest of this paper proceeds as follows. Section 4.1 proposes two approaches to making decisions about evaluating and selecting PPP projects. Section 4.2 provides a numerical illustration. Section 5 concludes this study.

### 4.1. Proposed Approaches

The multiple criteria for PPP projects’ evaluation and the selection problem are formulated as follows:

$A=\left(A_{1},A_{2},\ldots ,A_{m}\right)$: a set of m projects;

$C=\left(C_{1},C_{2},\ldots ,C_{n}\right)$: a set of n criteria;

$Y={\left[y_{ij}\right]}_{mn}$: the decision matrix.

Where $y_{ij}$ is the input data for the project i concerning measure j, i=1,2,…,m;j=1,2,…,n;

The decision matrix $Y={\left[y_{ij}\right]}_{mn}$ is normalized to the matrix $X={\left[x_{ij}\right]}_{mn}$ using the following formula:

$x_{ij}=\frac{y_{ij}-y_{j}^{\min}}{y_{j}^{\max}-y_{j}^{\min}}$, for benefit criteria;

$x_{ij}=\frac{y_{j}^{\max}-y_{ij}}{y_{j}^{\max}-y_{j}^{\min}}$, for cost criteria.

Where $y_{j}^{\max}=\max\left\{y_{1j},y_{2j},\ldots ,y_{mj}\right\}$ and $y_{j}^{\min}=\min\left\{y_{1j},y_{2j},\ldots ,y_{mj}\right\}$.

$W=\left(w_{1},w_{2},\ldots ,w_{n}\right)$: a set of criteria weights and $\sum_{j=1}^{n}w_{j}=1$.

Minimizing the total deviation from the best point is intuitively appealing. This approach’s principle is seeking to make all derived scores as close to the best point as possible. As discussed by Ma et al. (1999; 2020) [65,66], the matrix $X={\left[x_{ij}\right]}_{mn}$ is transformed into a weighted decision matrix $\Psi ={\left[z_{ij}\right]}_{mn}$, where
(1)$$z_{ij}=x_{ij}w_{j},i=1,2,\ldots ,m,j=1,2,\ldots ,n.$$

The best point is defined as $\Omega^{*}=\left\{z_{1}^{*},z_{2}^{*},\ldots ,z_{n}^{*}\right\}$, where
(2)$$\begin{array}{ll} z_{j}^{*} & =\max\left\{z_{1j}^{},z_{2j}^{},\ldots ,z_{mj}^{}\right\}\\ & =\max\left\{x_{1j}w_{j},x_{2j}w_{j},\ldots ,x_{mj}w_{j}\right\}\\ & =\max\left\{x_{1j},x_{2j},\ldots ,x_{mj}\right\}w_{j}\\ & =x_{j}^{*}w_{j}, \end{array}$$
and $x_{j}^{*}$ is the best value under the criterion j. It is straightforward because the maximal scores obtained from specific criteria are reasonably considered to reach [67]. The following function can measure the discrepancy between the performance under a specific criterion and the best point (Equation (3)):(3)$$\begin{array}{ll} D_{i} & =\sum_{j=1}^{n}{\left(z_{ij}-z_{j}^{*}\right)}^{2}\\ & =\sum_{j=1}^{n}{\left(x_{ij}-x_{j}^{*}\right)}^{2}w_{j}^{2}. \end{array}$$

A multi-objective programming model is reported below to optimize the performance of all projects (Equation (4)):(4)$$\{\begin{array}{c} \{\begin{array}{c} f_{1}=\min\sum_{j=1}^{n}{\left(x_{1j}-x_{j}^{*}\right)}^{2}w_{j}^{2}\\ f_{2}=\min\sum_{j=1}^{n}{\left(x_{2j}-x_{j}^{*}\right)}^{2}w_{j}^{2}\\ \ldots \\ f_{m}=\min\sum_{j=1}^{n}{\left(x_{mj}-x_{j}^{*}\right)}^{2}w_{j}^{2} \end{array}\\ s.t.\;\sum_{j=1}^{n}w_{j}=1, \end{array}$$
which is converted into a single-objective optimization model (Equation (5)):(5)$$\{\begin{array}{c} \min F=\sum_{i=1}^{m}\sum_{j=1}^{n}{\left(x_{ij}-x_{j}^{*}\right)}^{2}w_{j}^{2}\\ s.t.\sum_{j=1}^{n}w_{j}=1. \end{array}$$

To solve the quadratic programming (5), we construct a Lagrange function using a Lagrange multiplier λ (Equation (6)):(6)$$L=\sum_{i=1}^{m}\sum_{j=1}^{n}{\left(x_{ij}-x_{j}^{*}\right)}^{2}w_{j}^{2}+\lambda (\sum_{j=1}^{n}w_{j}-1)$$

The Hessian matrix of (6) concerning $w_{j}$ is diagonal, and its diagonal factors are $2\sum_{i=1}^{m}{\left(x_{ij}-x_{j}^{*}\right)}^{2}\geq 0$. Therefore, the Lagrange function has a minimum value, which is derived by differentiating (6) concerning $w_{j}$ and λ, respectively (Equation (7)):(7)$$\{\begin{array}{c} \lambda^{*}=\frac{1}{2\sum_{j=1}^{n}{\left[\sum_{i=1}^{m}{\left(x_{ij}-x_{j}^{*}\right)}^{2}\right]}^{-1}},\\ w_{j}^{*}=\frac{1}{\sum_{j=1}^{n}{\left[\sum_{i=1}^{m}{\left(x_{ij}-x_{j}^{*}\right)}^{2}\right]}^{-1}\sum_{i=1}^{m}{\left(x_{ij}-x_{j}^{*}\right)}^{2}}. \end{array}$$

Since the constraint of (5) is a non-empty convex set and the objective function is convex, the optimal solutions (7) to (5) are the optimal global solutions.

### 4.2. Numerical Illustration

To demonstrate the efficacy of the suggested approach, consider the following scenario: PPP projects are subject to various criteria, particularly indirect economic contribution ($y_{i1}$), direct financial contribution ($y_{i2}$), technical contribution ($y_{i3}$), social contribution ($y_{i4}$), and scientific contribution ($y_{i5}$). A summary of the input data is presented in Table 2.

Using Formula (7), the weights associated with different criteria are:(8)W=(0.1786,0.2487,0.2222,0.1521,0.1985)

Therefore, each project’s scores and rankings are reported in the seventh and eighth columns of Table 2 and Figure 7 below.

It is observed that our approach, namely, minimizing the total deviation from the best point, considers the direct economic contribution as the most important criterion and the social contribution as the least important one. Furthermore, our approach provides a complete ranking of all projects, reflecting the proposed approach’s discriminatory power.

PPP refers to innovative strategies utilized in the public sector to contract via the private party, which brings its ability and capital to deliver projects on time to the budget. Mutuality, the prevailing party retains the accountability to provide these services to the public in a path that has advantages and delivers economic improvement and development in the quality of life [68,69]. This study develops a new approach that minimizes the total deviation from the best point to determine each criterion’s weight. A numerical example is presented to demonstrate this approach’s effectiveness in providing a complete ranking of all projects.

## 5. Conclusions

This paper finds that, from a legal and contractual perspective, a significant number of the legal regimes of the LAC region and, consequently, of their PPP contracts, present at least one primary defect: the occurrence of an illegal event, such as the payment of bribes, triggers the possible application of general rules of administrative law that materialize in a zero-tolerance approach and that implies the suspension of large infrastructure projects and the paralysis of the PPP sector.

This situation has prompted some jurisdictions to put forward specific measures to address the problem. As previously noted, each measure’s real impact cannot be established just yet, as it is its application over time in specific cases that will determine its success or failure. Nonetheless, all present and future efforts of public authorities of any jurisdiction should focus on striking a precise balance that makes it possible to harmonize the different interests at play, which are widely diverse because they comprise numerous players. Therefore, on the one hand, the measures should guarantee the punishment of those responsible for corruption, the possibility of practical cooperation to put an end to impunity, the recovery of assets, and compensation for damages caused by the wrongful act. On the other hand, they should also ensure the continuity and conclusion of the projects, their proper operation and maintenance, job security, investment repayment, and the economy’s development.

All these complex objectives cannot be achieved if the only response is zero tolerance. Instead, they can be attained through an alternative integral approach that, instead of indiscriminately affecting all relations by declaring contracts null and void, prevents the just from paying for the sinners and allows projects with a positive socioeconomic impact to be implemented promptly. In light of this, measures should encourage the admission of one’s crimes and those committed by others and promote, through criminal court proceedings, the conviction of those responsible. A balanced approach does not contradict the principles that support a zero-tolerance policy, because it reinforces the idea that the sanction imposed is a true reflection of the moral concerns that motivate it, as Davis points out, by ensuring that the punishment fits the crime and falls on the person responsible [70,71] (The author suggests that the zero-tolerance approach seems misguided because it ignores efforts made to prevent and overcome acts of corruption committed by companies, as well as alternative ways in which companies and governments could help combat bribery in public procurement).

At this point, it is critical to emphasize that contract cancellation may be one of the many options available to the authorities, in which case an orderly exit of the responsible company and its rapid replacement must be achieved through legal, financial, and economic procedures that are clear, transparent, and predictable. Thus, mechanisms must be sought that allow for a quick replacement of the concessionaire or operator that is excluded from the works due to acts of corruption and an orderly transfer of the infrastructure, even in those exceptional cases in which the state is qualified to complete its construction and eventually execute its operation and maintenance.

Similarly, other complementary tools to the nullity of the contract must be adopted to constitute a set of available instruments aimed at attaining the balance of all the interests at stake. In comparative legislation, some precedents show alternative approaches. For example, as an expert in the World Duty-Free case, Lord Mustill explained that, according to British law, the party affected by an act of corruption is entitled to rescind the contract and prevent its enforcement in the future. However, this faculty is optional rather than mandatory, as the affected party may opt for waiving its right to rescind the contract and instead enforce it according to its terms (in this regard, see World Duty Free Company v. Republic of Kenya. Similarly, see [70]).

In the context of European Community law, Directive 2007/66/EC of the European Parliament and of the Council of 11 December 2007 addresses the issue of ineffectiveness of the award of public contracts. The Directive allows participants of specific procurement processes to contest infringements committed during the bidding processes using an independent and fast procedure [72]. In some instances, the competent body may choose to declare the contract’s ineffectiveness or maintain its effects if there are “*overriding reasons relating to a general benefit*” that so require it. According to the Directive, the “*economic interests directly connected with contract*” in question do not constitute overriding reasons related to a general interest, and the costs originated by the delay in the execution of the contract or those resulting from the launching of a new procurement procedure are cited as examples. Suppose the competent body decides to maintain the effects of the agreement. In that case, the declaration of ineffectiveness must be replaced by some alternative, proportional, and dissuasive sanction, such as the imposition of fines or the proportionate shortening of the contract term [73,74,75,76,77,78,79,80].

The adoption of measures aimed at preventing the paralysis of the infrastructure sector is of paramount importance for a country’s economy.

This is demonstrated by the completion of the projects in issue, as well as the continuation of investments, the provision of infrastructure funding, the solicitation of new tenders, and the preservation and creation of new jobs.

## Figures and Tables

**Figure 1 ijerph-20-01175-f001:**
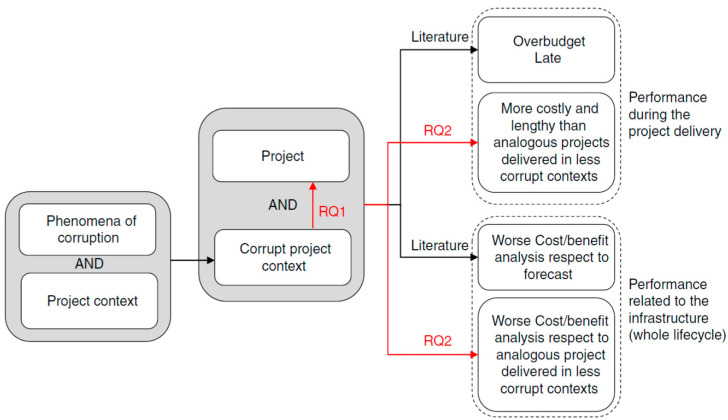
Presence of corruption in the PPPs. Source: Heydari et al. [7].

**Figure 2 ijerph-20-01175-f002:**
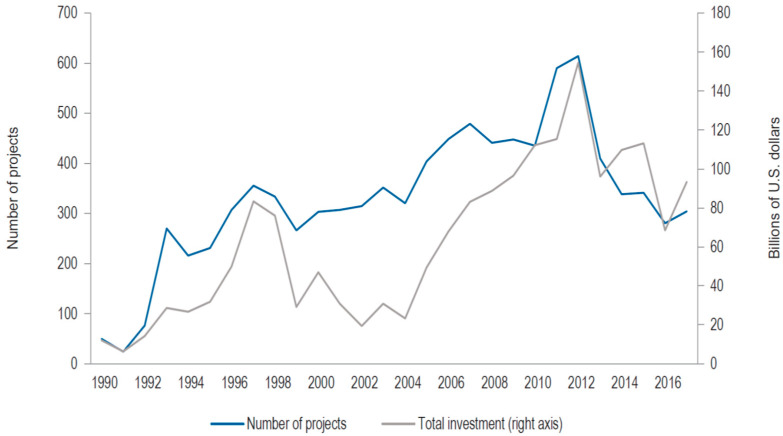
Number of projects and total investment in PPPs in emerging economies. Source: Heydari et al. [7].

**Figure 3 ijerph-20-01175-f003:**
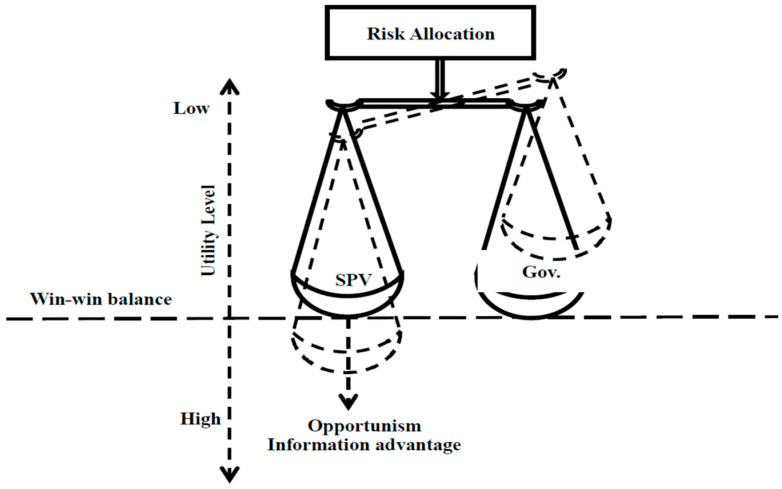
The role of risk allocation to achieve a win–win balance [7]. Note: SPV is special purpose vehicle and Gov is government.

**Figure 4 ijerph-20-01175-f004:**
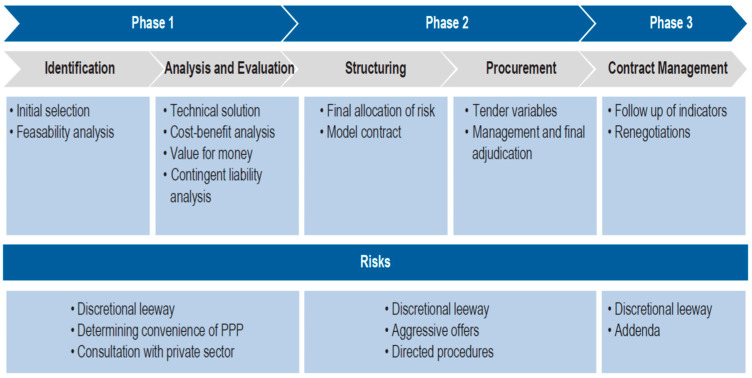
Phases of a PPP projects and risk of corruption. Source: Heydari et al. [7].

**Figure 5 ijerph-20-01175-f005:**
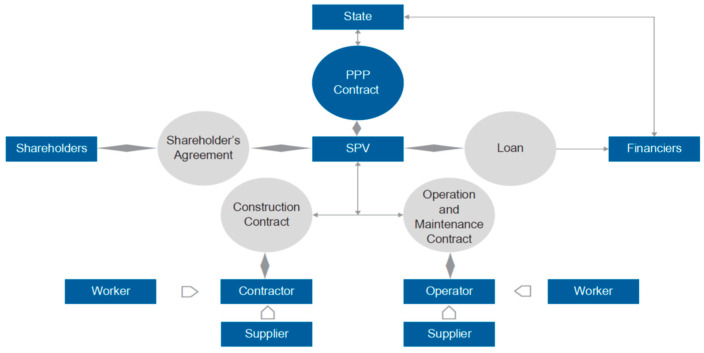
Primary contractual relations in a PPP contract. Source: Heydari et al. [7].

**Figure 6 ijerph-20-01175-f006:**
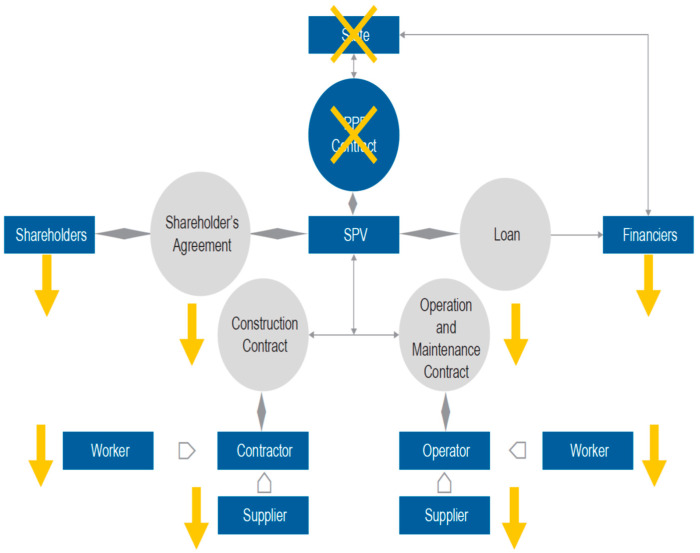
Contractual chain. Source: Heydari et al. [7].

**Figure 7 ijerph-20-01175-f007:**
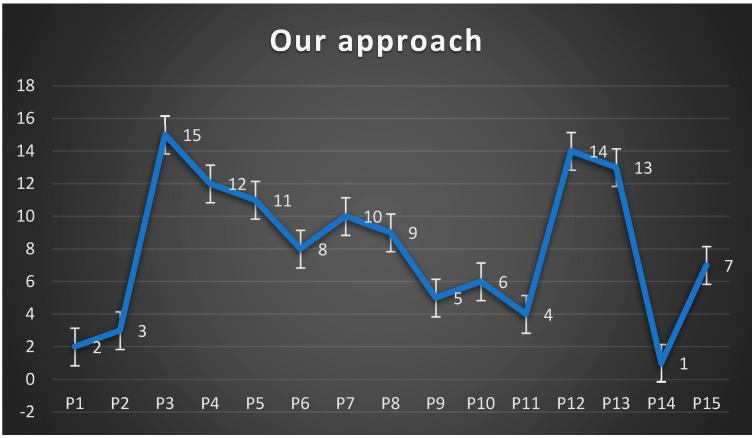
Ranking of PPP projects. Source: Heydari et al. [7].

**Table 1 ijerph-20-01175-t001:** Parameters and thresholds for megaproject evaluation.

Category	Parameter	Threshold
Cost	Cost escalation	>25%
Time	Slippage in execution times	>25%
Quality	Production versus the plan	Reduced production into year 2
Cost	Cost competitiveness	>25%
Time	Schedule competitiveness	>50%

Note: Due to the evaluation being too uneven due to the varying section lengths, the information regarding scheduling competitiveness is not examined [8].

**Table 2 ijerph-20-01175-t002:** Multiple criteria PPP projects.

Projects	Criteria					Scores	Ranking
	$y_{i1}$	$y_{i2}$	$y_{i3}$	$y_{i4}$	$y_{i5}$		
P1	67.53	70.82	62.64	44.91	46.28	0.8906	2
P2	58.94	62.86	57.47	42.84	45.64	0.8040	3
P3	22.27	9.68	6.73	10.99	5.92	0.0016	15
P4	47.32	47.05	21.75	20.82	19.64	0.3684	12
P5	48.96	48.48	34.9	32.73	26.21	0.5080	11
P6	58.88	77.16	35.42	29.11	26.08	0.6237	8
P7	50.1	58.2	36.12	32.46	18.9	0.5167	10
P8	47.46	49.54	46.89	24.54	36.35	0.5645	9
P9	55.26	61.09	38.93	47.71	29.47	0.6625	5
P10	52.4	55.09	53.45	19.52	46.57	0.6489	6
P11	55.13	55.54	55.13	23.36	46.31	0.6806	4
P12	32.09	34.04	33.57	10.6	29.36	0.3220	14
P13	27.49	39	34.51	21.25	25.74	0.3555	13
P14	77.17	83.35	60.01	41.37	51.91	0.9636	1
P15	72	68.32	25.84	36.64	25.84	0.6283	7

Source: Heydari et al. [7].

## Data Availability

The data presented in this study are available in [7,74]. Some of the data specifically about corruption is extracted from the unpublished doctoral dissertation titled “A cognitive basis perceived corruption and attitudes towards entrepreneurial intention” by Mohammad Heydari. The thesis was submitted to the School of Economics and Management, Nanjing University of Science and Technology, Nanjing, Jiangsu, China in 2020 to fulfill doctoral degree requirements in Management Science & Engineering [81].

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
