# Peer review of "Public Health Risk Evaluation through Mathematical Optimization in the Process of PPPs"

_ijerph, 2023, doi:10.3390/ijerph20021175_

Round 1

Reviewer 1 Report

I think that this article deals with an interesting topic and make a great contribution to the related field. However, I believe that (in some parts), by considering certain gaps and dispelling some existing issues, the research could result in higher quality than what is presented. To do so, the following comments and recommendations are provided:

1. Check all of the paper about formatting. And give a reference for each figure and table.

2. In all of the paper, wherever you have used (Eqs.), you need to cite it in the context accordingly.

3. Figures and tables should be cited accordingly in the context.

4. Since the authors are not native it is suggested that to check all of the paper about grammar etc.. and revise accordingly.

5. At the end, you must double-check your references; in some parts, information is missing.

6. Starting from pages (5, 9, 11, 12, 13) there are very long footnotes and it’s better to eliminate them or put them in the main text. 

7. Please state the academic contribution of the study, in particular, what research gap the authors intend to fill.

8. Based on the literature review in the paper, what research gap has been identified?

Author Response

12/27/2022

Dear respected editorial board;

We thank you for the valuable suggestions and comments of the reviews. We have revised the paper accordingly. We are sure that the paper has improved its quality remarkably.

  1. Check all of the paper about formatting. And give a reference for each figure and table.

All of the paper has been checked and for all of the fig(s) and Table(s) we give the references.

  1. In all of the paper, wherever you have used (Eqs.), you need to cite it in the context accordingly.

All Eq(s) were cited accordingly.

  1. Figures and tables should be cited accordingly in the context.

All of the fig(s) and table(s) were cited accordingly.

  1. Since the authors are not native it is suggested that to check all of the paper about grammar etc.. and revise accordingly.

All of the paper were checked and all of the grammar mistakes etc… were rewritten.

  1. At the end, you must double-check your references; in some parts, information is missing.

 All of the reference part rechecked and mistakes were solved.

  1. Starting from pages (5, 9, 11, 12, 13) there are very long footnotes and it’s better to eliminate them or put them in the main text. 

All of the long footnotes were added to the main body of the paper.

  1. Please state the academic contribution of the study, in particular, what research gap the authors intend to fill.

The detection of corrupt activities in public works has enhanced in recent years, partly thanks to the improved effectiveness of the controls implemented since the 1990s [37]. This new scenario requires an analysis of the economic effects of the existing legal responses for corrupt activities, particularly concerning the paralysis of the works and investments and the damages caused to third parties unrelated to the illicit act, including those interested in the project [38]. This paper is not intended to analyze the core causes of corruption in public works and the construction sector, which may derive from systemic or structural nature (economic informality and institutional weakness, among others).

  1. Based on the literature review in the paper, what research gap has been identified?

What is examined is how the primary legal mechanism to deal with corruption interplays with the specific characteristics of PPPs, bringing about particularly pernicious effects that have prompted some countries in the region to readjust their legal frameworks. On the other hand, the object of analysis is how the potential annulment of an ongoing contract triggers dire consequences in contractual structures based on structured financing and “contractual cascades” that combine public and private law. Figure 4 demonstrates a basic overview of the primary contractual relationships established in a typical PPP contract.

Reviewer 2 Report

Dear editor,

Thanks a lot for assigning the article to me for peer review. Although the topic of PPP is very important and can be in a great interest by health policymakers in a variety of health issues, the article suffers from adequate credibility and authenticity. It is not obvious which exact methodology is used to introduce different approaches of PPP evaluation. Was it achieved as a result of literature review or evidence synthesis? If yes, what are the RQ, Search Strategy, PRISMA FLOW chart, eligibility criteria, quality assessment appraisal tools and methods of analysis and synthesis. If no, from the introduction till the end of page 12 one may face with a pile of information which is not really integrated, synthesized and presented. The approach applied is more useful for a book chapter rather than an article. 

Starting from page 13, where the authors try to discuss the evaluating methods for PPP projects, it is again vague and ambiguous whether it is a comparative study to compare the pros and cons of different evaluation methods, if so it is not obvious which axis and items are compared and discussed and if no what is the rational and source of presenting these evaluation approaches? In this format the readers may face many questions about the reliability and validity of the study.  In other words, if it is a review, a description of the available approaches, a comparative study or an overview on the knowledge it should be clarified what is the exact methodology, protocol for collecting and presenting the information and the rational for final conclusion and synthesis based on the significance and application of PPP and the value of evaluation the projects in the health and public health sector.

Author Response

12/27/2022

Dear respected editorial board;

We thank you for the valuable suggestions and comments of the reviews. We have revised the paper accordingly. We are sure that the paper has improved its quality remarkably.

Was it achieved as a result of literature review or evidence synthesis? If yes, what are the RQ, Search Strategy, PRISMA FLOW chart, eligibility criteria, quality assessment appraisal tools and methods of analysis and synthesis. If no, from the introduction till the end of page 12 one may face with a pile of information which is not really integrated, synthesized and presented. The approach applied is more useful for a book chapter rather than an article. Starting from page 13, where the authors try to discuss the evaluating methods for PPP projects, it is again vague and ambiguous whether it is a comparative study to compare the pros and cons of different evaluation methods, if so it is not obvious which axis and items are compared and discussed and if no what is the rational and source of presenting these evaluation approaches? In this format the readers may face many questions about the reliability and validity of the study.  In other words, if it is a review, a description of the available approaches, a comparative study or an overview on the knowledge it should be clarified what is the exact methodology, protocol for collecting and presenting the information and the rational for final conclusion and synthesis based on the significance and application of PPP and the value of evaluation the projects in the health and public health sector.

The achieved results are based on literature review. As we mentioned:

What is examined is how the primary legal mechanism to deal with corruption interplays with the specific characteristics of PPPs, bringing about particularly pernicious effects that have prompted some countries in the region to readjust their legal frameworks. On the other hand, the object of analysis is how the potential annulment of an ongoing contract triggers dire consequences in contractual structures based on structured financing and “contractual cascades” that combine public and private law. Figure 4 demonstrates a basic overview of the primary contractual relationships established in a typical PPP contract.

Fig 4. Primary Contractual Relations in a PPP Contract

Source. Heydari et al. [14]

As we mentioned in the paper a multi-objective programming model is reported below to optimise the performance of all projects. To demonstrate the efficacy of the suggested approach, consider the following scenario: PPP projects are subject to various criteria, particularly, Indirect economic contribution (), Direct financial contribution (), Technical contribution (), Social contribution (), and Scientific contribution ().

It is observed that our approach, namely, minimising the total deviation from the best point, considers the direct economic contribution as the most important criterion and the social contribution as the least important one. Furthermore, our approach provides a complete ranking of all projects, reflecting the proposed approach's discriminatory power.

Round 2

Reviewer 2 Report

Dear authors,

Thanks for your response. Although you have mentioned that the present article is derived from a literature review, I still expect a more detailed description of the methodology in the article considering the following items:

- Careful consideration of the review type

- Your precise Research Question,

- The keywords, databases and search strategy

- approach of including the articles and determining their eligibility considering any quality appraisal tool and PRISMA flowchart

- approach of extracting the data, data analysis and synthesis 

As a guide please refer to the JBI guideline for the reviews. 

Warmest regards

Author Response

12/31/2022

Dear respected editorial board;

We thank you for the valuable suggestions and comments of the reviews. We have revised the paper accordingly. We are sure that the paper has improved its quality remarkably.

- Careful consideration of the review type

Systematic literature reviews (SLRs)

- Your precise Research Question,

This manuscript focuses on two research questions:

  • Q1: Which project characteristics favor corruption?

This first question is necessary to understand if there are attributes that make the projects more likely to suffer from corruption. The answer to this question is crucial, particularly for decision-makers and policy-makers in corrupt countries. For example, let us assume a “functional objective” (e.g. provide a certain amount of electricity in a certain area) that can be satisfied by two different projects, type, and A and B, and one of these (e.g. B) is more likely to attract corruption. Then, according to this criterion, A should be the correct choice in a corrupt project context.

  • Q2: How does a corrupt context affect project performance?

Since projects might have a poor schedule and budget performance even in “non-corrupt countries” this question aims to highlight the impact of corruption by comparing similar projects in different countries.

- The keywords, databases and search strategy

A critical literature analysis of sources, the majority outside the field of project management, is used to respond to question 1. Second question is addressed by the literature review and further investigated with a case study in (Section 4, and sub-section 4.1, and 4.2). First, the evaluation of the literature helps us to codify the major concepts that are important for this study, such as corruption, project context, corrupt project context, megaproject, etc. Second, it enables a list of key drivers indicating the kind of projects that are more likely to entail corruption to be included in the RQ1 response. Thirdly, it describes the impact corruption-related projects have on their performance over the course of their existence. In conclusion, the literature study emphasises two key points: the causes of corruption and how it affects project outcomes.

- approach of including the articles and determining their eligibility considering any quality appraisal tool and PRISMA flowchart

If on the one hand corruption is often particularly present in megaprojects, on the other hand, megaprojects are often associated with poor performance even in countries with low signals of corruption. Therefore, the methodology compares the megaprojects involved in the high-speed rail programs in Europe and globally.

This case is used as a reference because it is delivered in a corrupt project context and is technologically comparable to the other European high-speed rail programs [4]. The case study is designed to highlight the relationship between the endemic phenomenon of corruption and lower project performance; this approach is implemented according to the principles stated by [5-6]. The case study is made up of three main perspectives: the project context, the longitudinal view over the project lifecycle, and the transversal view about the project performance. The case study is designed in such a way for pragmatic reasons because (1) it is difficult to demonstrate directly the presence of corruption in projects and (2) the focus of the research is not on single corruption episodes, but the project delivered in a “corrupt project context”. Therefore, the manuscript indirectly shows the presence of corruption, by referring to the project context and by showing the results of the judicial processes and the investigations associated with the project. Fig. 1 summarizes the key research constructs and their casual interlinks along with the two RQs.

Fig 1. Presence of corruption in the PPPs

Source. Heydari et al. [7]

- approach of extracting the data, data analysis and synthesis 

The comparison considers two main perspectives; firstly, the project contexts and the extent to which it is corrupt. Secondly, the megaprojects performance, normalized, and adjusted to consider different environmental, urbanistic, and technological circumstances.

In particular, by adopting the research framework from [8], this manuscript shows how the project and project management performance evolves over the project lifecycle. Merrow's framework evaluates the megaproject performance through five parameters. Each parameter is associated with the threshold value, which permits to judge whether the performance is satisfactory or not (Table 1).

Table 1. Parameters and threshold for megaprojects evaluation

Category

Parameter

Threshold

Cost

Cost overruns

> 25%

Time

Slip in execution schedules

> 25%

Quality

Production versus plan

Reduced production into year 2

Cost

Cost competitiveness

> 25%

Time

Schedule competitiveness

> 50%

Note: The data about schedules competitiveness are not analyzed because the different length of sections makes the evaluation too uneven [8].
